# Digital Tools for Quantifying the Natural Capital Benefits of Agroforestry: A Review

**Stephen B. Stewart** [1,*], **Anthony P. O'Grady** [1], **Daniel S. Mendham** [2], **Greg S. Smith** [1] and **Philip J. Smethurst** [1]

1   CSIRO Land and Water, Sandy Bay, TAS 7005, Australia
2   CSIRO Land and Water, Canberra, ACT 2601, Australia
*   Correspondence: stephen.stewart@csiro.au

**Abstract:** Agroforestry is one nature-based solution that holds significant potential for improving the sustainability and resilience of agricultural systems. Quantifying these benefits is challenging in agroforestry systems, largely due to landscape complexity and the diversity of management approaches. Digital tools designed for agroforestry typically focus on timber and crop production, and not the broader range of benefits usually considered in assessments of ecosystem services and natural capital. The objectives of this review were to identify and evaluate digital tools that quantify natural capital benefits across eight themes applicable to agroforestry systems: timber production and carbon sequestration, agricultural production, microclimate, air quality, water management, biodiversity, pollination, and amenity. We identified and evaluated 63 tools, 9 of which were assessed in further detail using Australia as a case study. No single tool was best suited to quantify benefits across each theme, suggesting that multiple tools or models could be combined to address capability gaps. We find that model complexity, incorporation of spatial processes, accessibility, regional applicability, development speed and interoperability present significant challenges for the tools that were evaluated. We recommend that these challenges be considered as opportunities to develop new, and build upon existing, tools to enhance decision support in agroforestry systems.

**Keywords:** agroforestry; ecosystem services; natural capital benefits; nature-based solutions; decision support tools

## 1. Introduction

The concept of nature-based solutions is gaining increased traction within agricultural industries to address the global challenges of climate change, biodiversity loss and land degradation. A nature-based solution is one that aims to address sustainability challenges through the protection, sustainable management and restoration of both natural and modified ecosystems, benefiting both biodiversity and human wellbeing [1]. This approach contrasts with more traditional solutions that focus solely on enhancing yields of target products such as food or wood, because it also considers net ecosystem benefits over long periods. Approximately 47.8 million km$^2$ of the global land surface area (36.7% of total) is used for agriculture [2], and therefore the opportunity for nature-based solutions to improve sustainability is potentially very large. These opportunities are supported by a range of financial mechanisms such as certification schemes, green loans and bonds, and existing and emerging markets for carbon and biodiversity [3,4].

Agroforestry is one nature-based solution that holds considerable potential for addressing sustainability challenges in agricultural landscapes. Defined as the deliberate integration of woody perennial systems within the existing plant and/or livestock enterprise [5], agroforestry has been recognised for its potential to increase the resilience and profitability of agricultural systems [6–9]. Agroforestry builds natural capital on farms and thus can enhance the supply of provisioning, regulating and cultural ecosystem services (e.g., carbon sequestration, climate regulation, supply and diversification of

habitats [10–12]). These benefits have long been acknowledged qualitatively [13]; however, the availability of suitable quantitative data to support decision-making on the supply of these services under different configurations of species, farming systems and management goals is limited. This has presented a significant barrier to the broader adoption of agroforestry [14].

Despite these challenges, there has been a resurgence of interest in the use of agroforestry as a nature-based solution that can improve the resilience and sustainability of agricultural enterprises. This is supported by the recognised need for ethical and sustainable supply chains [15–17]. Accompanying this, there is a growing appetite for investment into nature-based solutions from private equity [18] that holds considerable potential to facilitate the adoption of agroforestry within the agricultural sector. Thus, understanding and quantifying the natural capital benefits generated from agroforestry systems is important in building the business case for adoption and unlocking the potential of emerging equity markets. However, this also means that there is a need for credible, feasible and accessible methods for quantifying these natural capital benefits.

The long history of research on agroforestry and ecosystem services has seen the emergence of many resources designed to address knowledge gaps, support decision-making, and build the case for the adoption of agroforestry [19,20]. These include knowledge databases, software packages, mathematical models, and guidance materials to support decision-making. The array of potential interactions and configurations in agroforestry systems, however, means that many of these resources can be difficult to apply in specific situations. Another challenge is that many of these resources are no longer actively maintained, and therefore may not be accessible to land managers. Tools such as WaN-uLCAS [21], Hi-SAFE [22], SCUAF [23] and HyPAR [24] have been developed to quantify tree-crop interactions at field scale using process-based models; however, these can be difficult to implement in practice due to their complexity. The Agricultural Production Systems sIMulator APSIM; [25,26] is a modular modelling framework that been suggested as one option for simplifying and overcoming this complexity [27,28], and it has been applied to agroforestry [29,30]. The Imagine tool [31] was developed to understand the effects of trees on the economics of agricultural systems and has the flexibility to bring in other aspects of natural capital (Mendham, 2018). Despite the availability and utility of these existing tools, there are many more resources available in the ecosystem services modelling domain that can quantify the broader natural capital benefits of agroforestry systems [7].

Many tools have been developed to model ecosystem services, quantify pools of natural capital, and provide standardised methodologies in support of market-based offsetting schemes. While some of these tools comprise methods libraries and guidance materials (e.g., https://cdm.unfccc.int/, accessed 13 August 2022, TESSA; [32]) that require the user to implement their own technical workflows, many are distributed via software packages or digital platforms. Two prominent examples from the ecosystem services modelling and natural capital accounting literature are ARIES (ARtificial Intelligence for Ecosystem Services; [33]) and InVEST (Integrated Valuation of Ecosystem Services and Tradeoffs; [34]). These tools bring together an assortment of ecosystem services models and have been applied from local to national scales [35–39]. Others have focused more strongly on carbon storage and sequestration services in support of national reporting requirements and carbon offsetting mechanisms (e.g., FullCAM, FLINT; [40,41]). These tools are useful for quantifying natural capital; however, they are not always fit-for-purpose [42] and many have historically required significant expertise and time to operate [43]. These challenges have been broadly recognised and have driven the ongoing development of desktop software and APIs, web platforms (e.g., ARIES for SEEA explorer, https://aries.integratedmodelling.org, accessed 1 September 2021; LOOC-C, https://looc-c.farm/, accessed 15 September 2021; FlintPro, https://flintpro.com/, accessed 1 October 2021), and data repositories that lower the resources required to produce quantitative estimates of ecosystem services.

Despite the wide range of tools that have been developed to quantify the various elements of natural capital in agroforestry systems, it can be difficult to determine which are best fit for purpose. Existing tools have often been designed for specific regions of the world, with different target spatial scales, data requirements and modelling capabilities. Agroforestry is practiced in complex managed systems, often with a small spatial footprint (e.g., shelterbelts), and therefore spatial scale and configuration are particularly important to represent the variety of land uses and interactions that may occur. Many of the tools specifically designed for modelling agroforestry systems are focused on biophysical processes (e.g., water balance, uptake of soil nutrients, productivity) or are narrow in scope (e.g., specific crop types), and therefore the range of natural capital benefits are not well represented [28]. The objectives of this review were to:

1.  Identify tools that quantify natural capital benefits of agroforestry and shortlist those best suited to farm-scale applications in Australia;
2.  Evaluate the modelling capabilities of the shortlisted tools; and
3.  Identify key capability gaps and opportunities for future development.

We first consider a broad range of the available tools to evaluate existing international capabilities but focus in detail on Australia as a case study to reflect the need for locally applicable models. The strengths and limitations of each approach to quantifying the natural capital benefits of agroforestry are evaluated, followed by a discussion of the key findings and recommendations for future development.

## 2. Methods

### 2.1. Review Scope

The natural capital approach extends the economic notion of capital (resources that enable economic production) to the natural environment. The term 'natural capital' conceptualises nature as assets: stocks of resources such as clean air, water, soil and living things which produce flows of ecosystem services that have value because they benefit humans (households or firms). For clarity, in this paper, we use the term 'natural capital benefits' (NCBs) to refer to these benefits.

A distinction was made between digital tools or platforms, stand-alone datasets, bespoke mathematical models, and guidance materials in identifying the tools to include in this review. This distinction was made to constrain the scope of analyses and to focus on tools that facilitate the computation of agroforestry NCBs. Bespoke mathematical models or analyses that are found in the scientific literature were excluded as they are often calibrated for specific applications or case studies and often require significant effort to reproduce. Guidance materials (e.g., pdf tools and instructional videos) were also excluded because they do not allow for the direct computation of natural capital benefits. Stand-alone datasets were not considered as they are a result, rather than an implementation, of applied methods. By the term tool, we refer to software and platforms that provide access to digital implementations of models or functions that can be used to quantify biophysical, economic, and social or cultural variables of interest.

The capabilities of each tool were evaluated on eight key themes based on our understanding of agroforestry systems [7,27,28]. These were selected to reflect existing modelling capabilities and the importance of different natural capital benefits in agroforestry systems. These themes were:

1.  Timber production and carbon sequestration;
2.  Crop, pasture, and livestock production;
3.  Wind, shelter, and microclimate;
4.  Air quality and pollution;
5.  Erosion, runoff, and flood mitigation;
6.  Biodiversity;
7.  Crop pollination; and
8.  Amenity and recreation.

This review was designed to capture a broad range of tools that can quantify the NCBs of agroforestry systems. Tools designed specifically for agroforestry typically focus on timber and food production, with mechanisms to model tree-agriculture interactions. While these tools have been considered, detailed reviews of these tools are also provided by Luedeling et al. [27] and Kraft et al. [28]. Both reviews note the limited ability of existing agroforestry tools to quantify the broader range of NCBs. These recognised limitations were a significant factor in reviewing such a wide range of potential tools.

### 2.2. Identifying Tools That Quantify Natural Capital Benefits of Agroforestry

Tools were identified using web search engines supplemented by recommendations from the authorship team. Search terms were constructed by combining each of the NCBs with 'agroforestry', 'tool', 'calculator', and 'platform'. The terms 'natural capital' and 'ecosystem services' were also used in conjunction with the NCB descriptions to identify many relevant tools. All web searches were conducted from September to October 2021. Many of the tools that were identified are frequently updated or modified, therefore all subsequent tool evaluations did not consider software changes after these dates. Tools were not considered where sufficient information could not be found, for example where web links were no longer functional.

Each of the identified tools was assessed to determine which of the eight key NCB themes listed above could be quantified. The evaluation was performed by either downloading and testing the software, reading technical manuals or scientific manuscripts, or by assessing the information that could be found online at the websites for each respective tool. Any economic measures were also noted, as these were common features of many tools.

### 2.3. Shortlisting Tools Best Suited to Farm-Scale Agroforestry Applications in Australia

The list of tools was screened to identify those likely to be most effective in quantifying NCBs at the farm scale in Australia. Tools were considered suitable for farm-scale analyses where outputs could be either directly (e.g., high precision mapping) or indirectly (e.g., area-weighted scaling) attributed to specific landscape features (e.g., shelterbelts, paddocks). This shortlist was created by process of elimination using a series of six criteria. Tools were included in this shortlist where they:

1. Quantify at least one of the selected natural capital benefits of agroforestry;
2. Use methods or data that are compatible with farm-scale analyses;
3. Include, or allow the user to provide, supporting datasets (where required) that can be applied in Australia;
4. Use methods and models that are suitable for Australian applications, or can be applied without extensive parameterisation;
5. Provide functionality not implemented by existing Australian tools; and
6. Are currently available for use as open-source software, via collaboration, or by web application.

Each of the tools meeting these criteria were selected for further evaluation of modelling capabilities.

### 2.4. Evaluating the Modelling Capabilities of Shortlisted Tools

Each of the eight NCB themes was assessed in further detail for the shortlisted tools. The name of the model (or tool module), a short description of the methodological approach, and a summary of the key outputs that are quantified were described. Models were described separately where a shortlisted tool provided multiple options for quantifying one of the NCBs. Where available, the economic valuation capabilities of the shortlisted tools were discussed for each NCB. The strengths and limitations of each approach is briefly discussed, followed by a summary of potential alternative approaches that are present in other tools or the scientific literature.

## 3. Results

*3.1. Identifying Tools That Quantify Natural Capital Benefits of Agroforestry and Shortlisting Those Best Suited to Farm-Scale Applications in Australia*

A total of 63 candidate tools were identified and evaluated (Table A1), 9 of which were shortlisted after meeting the required criteria (Figure 1; Table 1). Timber production and carbon sequestration were the most frequently included natural capital benefits and were quantified by 62% of identified tools and 89% of those shortlisted. The biodiversity, erosion, runoff, and flood mitigation NCBs were the next most common, included by 40% and 32% of identified tools, respectively. Methods for providing an economic valuation of NCBs were common across all identified (48%) and shortlisted (78%) tools; however, this did not mean that methodologies had been implemented for all NCBs modelled by each tool. For example, carbon and timber production was often valued where biodiversity was not.

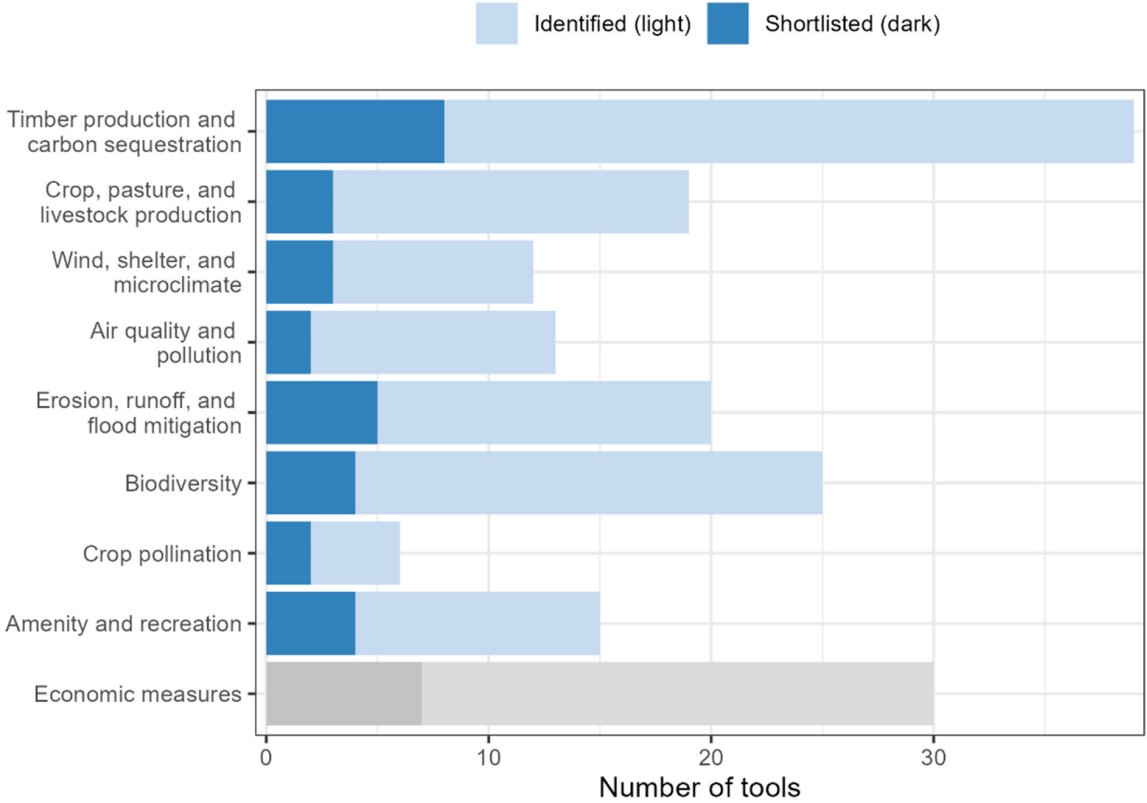

**Figure 1.** Total number of identified and shortlisted tools that quantify each of the selected natural capital benefits of agroforestry, in addition to those with the capability for calculating economic measures.

**Table 1.** Summary of shortlisted tools that can quantify the potential natural capital benefits of agroforestry.

| Tool | Software Type | Spatial Analysis Type | Timber Production and Carbon Sequestration | Crop, Pasture, and Livestock Productivity | Wind, Shelter, Land Microclimate | Air Quality and Pollution | Erosion, Runoff, and Flood Mitigation | Biodiversity | Crop Pollination | Amenity and Recreation | Economic Measures | Citation/Link |
|---|---|---|---|---|---|---|---|---|---|---|---|---|
| APSIM [A] | Desktop app, command line | Point or area | x | x | x | x | x | | | | x | [25,26], https://www.apsim.info/ |
| ARIES (for SEEA explorer) [B] | Web app, k-LAB software | AI-selected grid | x | | | | x | x | x | x | x | [33], https://aries.integratedmodelling.org/ |
| Farm Forestry Toolbox | Desktop app | Point or area | x | | | | | | | | x | [44], https://www.farmforestrytoolbox.com/ |
| FullCAM 2020 | Desktop app, command line | Point or area | x | x | | | | | | | x | [41], https://www.industry.gov.au/data-and-publications/full-carbon-accounting-model-fullcam |
| Imagine | Desktop app | Point or area | x | x | x | | | | | x | x | [31,45] |
| InVEST | Desktop app, python API | User-selected grid and/or area | x | | | | x | x | x | x | x | [34], https://naturalcapitalproject.stanford.edu/software/invest |
| i-Tree Eco | Desktop app | Point or area | x | | x | x | x | x | | | x | [46], https://www.itreetools.org/tools/i-tree-eco |
| LUCI | ArcGIS plugin | User-selected grid and/or area | x | | | | x | x | | | | [47,48], https://lucitools.org/ |
| SolVES | ArcGIS/QGIS plugin | User-selected grid | | | | | | | | x | | [49], https://pubs.er.usgs.gov/publication/tm7C25 |

Notes: [A] Two versions of APSIM were considered, as not all modules developed for v7.1 [26] have been implemented in NextGen [25]. [B] ARIES was evaluated using 'ARIES for SEEA explorer' as it is the most recent available implementation of the tool and is highly accessible.

Each of the 9 shortlisted tools and a summary of their features are described in Table 1. Of these tools, ARIES, InVEST, LUCI and SolVES are designed to operate spatially and produce spatial outputs (e.g., gridded raster data, polygons). For ARIES, we focused on the ARIES for SEEA explorer web application due to the increased accessibility of the tool, though notionally users can provide their own data. InVEST includes a broad range of well-documented ecosystem services models that are similar in scope to the ARIES offerings, though vary in their implementation. LUCI is designed to model ecosystem services and identify optimal land use configurations at very high resolution (i.e., 5 m pixels). It has a strong focus on hydrology, erosion, and nutrient transport. SolVES is unique in that it focuses specifically on statistically linking social values, including amenity and recreation, to spatially explicit environmental layers. The remainder of the shortlisted models produce outputs that are representative of a single point or area; however, can typically be run in batch mode or otherwise post-processed to form a spatially continuous output. FullCAM is used to calculate Australia's greenhouse gas emissions in the land sector and allows users to calculate carbon storage and sequestration under a wide range of management and disturbance scenarios. APSIM is a modular framework for modelling agricultural systems and has been demonstrated in agroforestry applications [29,30]. The i-Tree suite of tools rely heavily on datasets that are only currently available for North America. We focused on i-Tree Eco as it includes several unique approaches to relevant NCBs and can be applied internationally. The remaining two tools, the Farm Forestry Toolbox and Imagine, have a much stronger focus on economic performance and trade-offs. While the Farm Forestry Toolbox is focused on timber production, Imagine considers the broader farming system. Imagine has a suite of its own algorithms for predicting crop, pasture, and livestock growth as well as interactions between alternative land uses (based on climate), and it can also accept input from more detailed models such as APSIM and GrassGro [50].

There were 54 tools that did not meet each of the shortlisting criteria. The frequencies of criteria not met across all identified tools are shown in Figure 2. The natural capital benefits of agroforestry were not adequately quantified by 32% of tools (a). These included the Atlas of Living Australia and Integrated Biodiversity Assessment Tool, that despite both describing biodiversity assets, do not quantify their relationship with agroforestry. Many tools also rely upon datasets that are not available for Australia (c, 21%), or models that are not suitable for Australian applications or cannot be applied without significant parameterisaton (d, 25%). For example, many of the tools have been designed for use in specific regions such as North America (e.g., i-Tree Design, i-Tree Landscape, InForest) or the United Kingdom (e.g., Pollution removal by vegetation, B£ST, EFISCEN, Greenkeeper) and rely upon regionally specific datasets and models. The typical spatial resolution of analysis was a limiting factor for many tools that use gridded input data; however, it was the dominant reason for exclusion for only two models (b, LUTO and Co$ting Nature). Several of the tools that were ultimately shortlisted are usually run at a resolution that is too coarse for farm-scale analyses (e.g., 250 m grid cells); however this is offset by the ability to provide user-defined data (e.g., ARIES, InVEST) or apply area-weighted scaling (e.g., APSIM, FullCAM 2020) for farm-scale applications.

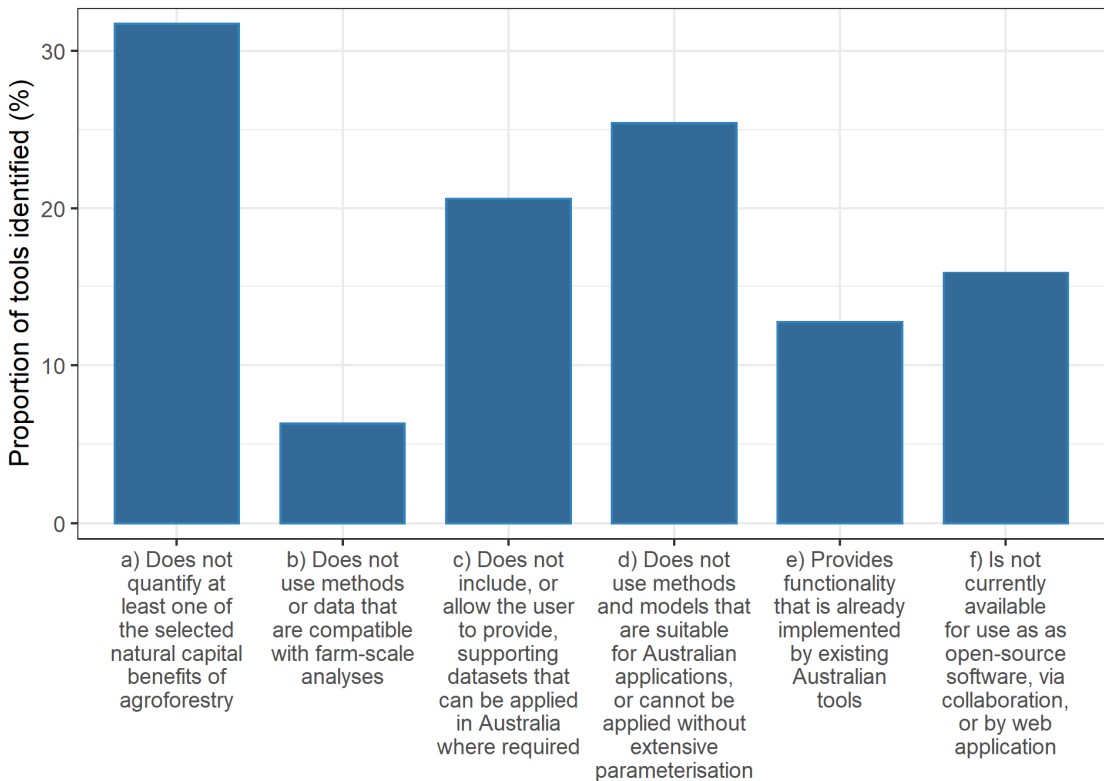

**Figure 2.** Proportion of tools identified (*n* = 63) that did not meet the respective suitability criteria for detailed evaluation of modelling capabilities.

### 3.2. Evaluating the Modelling Capabilities of Shortlisted Tools

3.2.1. Timber Production and Carbon Sequestration

All but one (SolVES) of the shortlisted tools quantify timber and/or carbon sequestration, though with methods of varying complexity (Table 2). The models range from simple methods that link land cover to carbon stocks via lookup tables (ARIES, InVEST, LUCI), to biophysical models that track the storage and transport of carbon between the soil, atmosphere, vegetation (branches, bark, twigs, leaves and roots), debris and harvested products (i.e., 'pools'). The biophysical models CABALA (when used externally by Imagine), the Farm Forestry Toolbox and APSIM model stand volumes, enabling timber production estimates. The potential for timber sales, trade of carbon offsets and the social cost of carbon mean that economic measures are commonly included in these tools.

Timber production and carbon sequestration are among the most tangible NCBs that can be provided by agroforestry. The models described above represent an array of different approaches that have strong foundations in both policy instruments and scientific literature. They enable estimates of carbon storage and sequestration in data-poor environments (e.g., lookup tables), but also include much more complex empirical and process-based biophysical models that explicitly incorporate management and disturbance regimes. The lookup table-based approach is limited in that it relies heavily on mean values (typically without consideration for spatial variations in productivity) and cannot track fluxes between carbon pools over time. Empirical and biophysical models also variously rely upon many assumptions, and the importance of site-specific conditions (e.g., soil properties, groundwater access, climatic variability, pests and disease, species) means that local monitoring and validation can be required to verify expectations.

**Table 2.** Summary of the shortlisted tools and associated models that quantify timber production and carbon sequestration.

| Tool | Model Name | Description | Relevant Output Variables |
|---|---|---|---|
| APSIM | *Eucalyptus, Pinus, Gliricidia* | Biophysical tree models calibrated for specific genotypes within each genus. | Biomass and carbon (soil, aboveground and belowground plant pools); stem diameter, height, and volume. |
| ARIES | Carbon storage | Carbon stocks are estimated by linking land cover/use, ecofloristic regions, continents, forest cover, and fire history to lookup tables following Ruesch and Gibbs [51]. Soil carbon stocks are taken from the global ISRIC database. | Carbon (total stored in aboveground vegetation, belowground vegetation, first 200 cm of soil). |
| Farm Forestry Toolbox | Site productivity | Annual indices of site productivity, integrated with species-specific empirical growth models. | Stand volume, mean annual increment, basal area, mean dominant height, diameter at breast height. |
| FullCAM 2020 | FullCAM | The Full Carbon Accounting Model [41] is used for modelling Australia's national greenhouse gas emissions in the land sector. It is used to estimate carbon stocks, sequestration and emissions associated with vegetation and soil. | Biomass and carbon storage (aboveground and belowground by pool), emissions (decomposition, fire). |
| Imagine | CArbon BALAnce (CABALA) | A dynamic forest growth model that incorporates water, carbon and nutrient balances designed for decision support in silvicultural systems [52]. | Biomass (aboveground and belowground by pool), stand volume, height, basal area, diameter at breast height. |
| InVEST | Carbon storage and sequestration | Carbon storage and sequestration are estimated using a land cover/use lookup table. Sequestration is calculated using linear interpolation where a future scenario is available. | Carbon (total stored and sequestered in aboveground vegetation, belowground vegetation, soil, and dead biomass) |
| i-Tree Eco | Carbon storage and sequestration | Models carbon stocks and sequestration rates by applying allometric equations to tree structural characteristics. | Carbon (total stored, sequestered annually, and emitted due to decomposition) |
| LUCI | Carbon stocks and fluxes | Carbon storage in biomass and soil is estimated as steady state for different combinations of and land cover/use classes and soil type. Net emissions or sequestration based on alternative scenarios. | Carbon (total stored in biomass and first 30–100 cm of soil, net emissions, or sequestration) |

Notes: Imagine includes a simple tree growth algorithm. It is not described here as there are more suitable options such as CABALA for use in agroforestry contexts. CABALA outputs are imported into Imagine [45] and must be run separately.

The long history of research into silviculture and the distribution and density of global biomass has supported the development of many models. International initiatives such as the United Nations Programme on Reducing Emissions from Deforestation and Forest Degradation (UN-REDD) and Intergovernmental Panel on Climate Change (IPCC) Task Force on National Greenhouse Gas Inventories (TFI) have also formalised a hierarchy of methodologies for carbon accounting in the land sector. The process-based 3-PG model [53], used to estimate site productivity indices required by FullCAM, has been widely applied in Australia and internationally for modelling forest growth [54–56]. LOOC-C (https://looc-c.farm/, accessed 15 September 2021) is a user-friendly web-application designed to rapidly identify carbon abatement project opportunities under Emissions Reduction Fund (ERF) and Land Restoration Fund (LRF) in Australia. It rapidly provides users an estimate of emissions reductions for different project activities across Australia; however,

it is currently focused on soil carbon, livestock, and regeneration of native vegetation. Remotely sensed estimates of biomass are also increasingly becoming available [57–61]; however, the relative error of these products can be high and they describe current or past conditions, so therefore alone are of limited use in evaluating alternative agroforestry configurations.

### 3.2.2. Crop, Pasture, and Livestock Production

Crop, pasture, and livestock production can be modelled in Australian agroforestry systems by three of the shortlisted tools (Table 3). APSIM is widely used in Australia and internationally for modelling crop biomass and food production [25] and can also be used to predict pasture and livestock productivity. FullCAM includes modules for modelling crops, pasture, and grazing-based emissions; however, it does not explicitly allow for tree-crop interactions. Imagine includes several integrated options for estimating crop, pasture, and livestock productivity, and can incorporate predictions from more complex models. Both APSIM and Imagine allow for competition and interactions between trees, crops, and pasture, and therefore are well suited to agroforestry modelling.

**Table 3.** Summary of the shortlisted tools and associated models that quantify crop, pasture, or livestock production.

| Tool | Model Name | Description | Relevant Output Variables |
|------|-----------|-------------|---------------------------|
| APSIM | Active tree in strip crop system, using *Eucalyptus*, *Pinus* or *Gliricidia* | Simulates competition between one tree zone and one crop zone; several tree, crop, pasture and livestock options. | Crop, pasture, or livestock production |
| APSIM | Tree proxy in multi-zone system | Simulates competition with user-defined tree characteristics; multiple tree and crop zones; several crop, pasture, and livestock options. | Crop, pasture, or livestock production |
| FullCAM | Various options for crops, pastures, and grazing | The Full Carbon Accounting Model [42] is used for modelling Australia's national greenhouse gas emissions in the land sector. It is used to estimate carbon stocks, sequestration and emissions associated with vegetation and soil. | Biomass and carbon storage (aboveground and belowground by pool), emissions (decomposition, fire). |
| Imagine | N/A | Simple, integrated crop, pasture, and livestock production models that simulate spatial-temporal competition via yield factor adjustments. | Crop, pasture, or livestock production |

Notes: Imagine is designed to allow external inputs, and therefore more advanced models such as GrassGro [50] and APSIM may be used in place of the integrated models.

Imagine allows for spatial interactions via yield-based adjustment factors that can be modified for water availability. This approach is flexible in that it can be used to incorporate productivity modifiers without the need for running complex biophysical models. These interactions can be simulated in APSIM using two key approaches. The first approach simulates productivity in adjacent tree and agriculture zones. It accounts for competition for light, water and nitrogen using climate, soil, and management as key inputs [62], and can simultaneously predict timber production for the tree species selected. In the second approach, the competitive impacts of trees are simulated by defining the behaviour of a generic tree proxy. Competition for light, water and nitrogen is then used to constrain pasture and crop productivity [30,63]. Multiple tree and agricultural zones are allowed when using tree proxies; however, timber production is not simulated.

APSIM is highly modular and has been used to simulate a range of tree-crop transects in Australia and internationally. The current range of tree, crop, pasture, and livestock types does, however, remain somewhat limited. While contributions to the development of APSIM are welcomed, the development and calibration of new models can be resource intensive. Functionally similar plant or livestock types may be suitable where dedicated modules have not yet been developed. The integrated productivity functions that are

provided with Imagine are limited in comparison to more complex biophysical models like APSIM. This is offset by the ability to import predictions from secondary models.

Many models and tools have been developed to predict crop, pasture, and/or livestock production. GrassGro is an alternative pasture and livestock production modelling tool that is commonly used in Australian beef and sheep grazing systems [50]. AussieGRASS [64] simulates pasture growth across Australia, quantifying biomass, grass curing and fire risk, among other variables. DSSAT [65] shares similarities with APSIM and is used internationally for crop and pasture modelling, but it does not currently support trees or agroforestry options. STICS [66] is also much like APSIM and includes agroforestry options; however, it is not used widely in Australia as it is designed for European applications. Comprehensive reviews of tools developed for process-based modelling of agroforestry systems are provided by Luedeling et al. [27] and Kraft et al. [28], many of which are briefly described in Table A1.

### 3.2.3. Wind, Shelter, and Microclimate

The impacts of trees on wind, shelter, and microclimate could be quantified by three shortlisted tools (Table 4). Microclimate here refers to weather experienced locally as affected by trees, i.e., underneath and adjacent to trees at different distances across a farm. Each tool quantifies a different aspect of this potential NCB. APSIM includes three key modules that are typically applied in sequence. APSIM AgroforestrySystem estimates relative changes in wind speed following [29] in a transect of zones away from a tree row, which are then used in the next module (APSIM LocalMicroClimate) to simulate the corresponding impacts on weather variables such as evapotranspiration within a zone. APSIM MicroClimate then computes water and energy balance characteristics across multiple competing canopies within a zone; however, it assumes horizontally uniform canopy layers. Imagine draws directly upon experimental evidence to relate the characteristics of shelterbelts to changes in livestock mortality [14] and pasture productivity [45]. The impact of tree cover on shade availability and ultraviolet radiation (UV) is quantified by i-Tree Eco based on [67]; however, this is limited in that it is based on the average vegetation characteristics across an area and has been designed for describing impacts on human health in urban environments.

**Table 4.** Summary of the shortlisted tools and associated models that quantify wind, shelter, and microclimate.

| Tool | Model Name | Description | Output Variables |
|---|---|---|---|
| APSIM | AgroforestrySystem, LocalMicroClimate and MicroClimate | Calculates weather inputs to each zone, and the energy and water balance parameters across competing canopies within each zone [29]. | Proportional reduction in wind speed. Rainfall and radiation interception, canopy conductance, potential transpiration. |
| Imagine | N/A | Empirical adjustment of livestock mortality [14] and pasture production, based on distance from tree belt in units of tree heights; [45]. | External feed requirement of a fixed herd based on change in pasture production, reduction in livestock mortality. |
| i-Tree Eco | Ultraviolet radiation | Estimates the reduction in ultraviolet radiation provided by tree shade across an area based on [67] using vegetation characteristics and weather data. | Protection factor, reduction in UV index, percent reduction, overall UV index, shaded UV index |

The complexity of interactions between the structural attributes of tree cover and wind, shelter, and microclimate means that these NCBs are difficult to quantify with confidence [14]. Each of the approaches implemented by the shortlisted tools are therefore simple by necessity, despite having been developed based on credible scientific studies. This simplicity can be an advantage where the assumptions and limitations are made explicit. For example, the empirical adjustment method applied with Imagine acts directly on the final service (i.e., change in pasture production and livestock mortality), allowing

for the direct calculation of gross margins and net present value across the farming system. There are, however, many potential factors to consider that can make such approaches difficult to generalise. None of these methods explicitly consider the spatial configuration and porosity of trees, the spatial and temporal variations in wind speed and direction, or corresponding impacts on temperature and humidity, each of which are important factors in determining how any potential benefits are realised [68–73].

Several alternative methods have been applied by other tools and in the scientific literature. EcoServ-GIS [74] applies simple buffers to trees and greenspaces of varying patch size to quantify local climate services in the United Kingdom, largely based on studies conducted in urban environments. Field-scale agroforestry modelling frameworks such as WaNuLCAS [21], Hi-SAFE [22], and HyPAR [75] allow for canopy interactions and the various associated impacts on wind, shelter and microclimate; however, their complexity and extensive parameterisation requirements can present significant barriers to implementation (see [27]). The TOPEX index [76] characterises topographic exposure using digital elevation models and is used for forest and windthrow risk assessments [77–79], though does not directly quantify the benefits that shelterbelts may provide. The wind chill index [80,81] has been associated with risk of lamb mortality, and shelterbelts best offset this risk in locations where wind speed drives the calculation more strongly than temperature or precipitation [72].

### 3.2.4. Air Quality and Pollution

Of the shortlisted tools, only i-Tree Eco and APSIM quantify effects on air quality and pollution (Table 5). Three models are available in i-Tree Eco. The first estimates the change in concentration of air pollutants by dry deposition on vegetation following Nowak et al. [82] and can be used to draw direct links between trees and human health impacts [83,84]. The second model estimates oxygen production by trees though this is likely to be negligible given the large amount of oxygen present in the atmosphere, significant contributions of algae and other photosynthetic organisms that inhabit marine ecosystems to global oxygen production (conservatively over 50%; [85]), and comparatively small scale of typical agroforestry plantings (e.g., shelterbelts). The final model estimates the production of biogenic volatile organic compounds (BVOCs) by trees (see [86]) that can be indirectly linked to impacts on human health as precursors to ozone and secondary organic aerosols [87–89].

APSIM simulates nitrous oxide ($N_2O$) production, an important greenhouse gas. Simulated production of $N_2O$ has been tested in several agricultural contexts [90]. Reliable predictions require local calibration and potential improvements have been identified [91]. The main purpose of simulating $N_2O$ in APSIM is to maintain the N balance of the simulated system and to quantify losses by this mechanism.

**Table 5.** Summary of the shortlisted tools and associated models that quantify air quality and pollution.

| Tool | Model Name | Description | Output Variables |
|------|------------|-------------|------------------|
| APSIM | Nutrient | Daily emissions-based soil N, water, and temperature. | Mass of $N_2O$-N released. |
| i-Tree Eco | Air pollution removal | Estimated air pollution (CO, $NO_2$, $O_3$, $SO_2$, $PM_{10}$ and $PM_{2.5}$) removed by grasses, shrubs, and trees. Model requires species, vegetation cover, and both weather and pollutant observations. | Change in pollutant concentration. |
| i-Tree Eco | Oxygen production | Estimated net oxygen production as a proportion of net carbon sequestration. | Net oxygen produced. |
| i-Tree Eco | Volatile Organic Compound (VOCs) emissions | Estimated biogenic volatile organic compounds (BVOCs; isoprene and monoterpene) emitted by trees. Model requires species, height (total and crown to base) crown width and percentage canopy missing. | Isoprene and monoterpene emissions. |

The proportion of air pollutants that are removed by vegetation is typically less than 1% [82,92], and it can be overestimated by i-Tree [93,94]. The air pollution removal model relies upon point-source data for both weather and pollutant concentrations. These data vary in availability, both spatially and temporally, across different parts of the world. Furthermore, they do not consider the contribution of BVOC emissions to the production of ozone and secondary organic aerosols. Despite these limitations, an advantage of the air pollution removal model is that the results can be associated with impacts on human health (or externality costs not captured by the source processes), and therefore a financial value can be applied to the benefits of trees. The valuation provided by i-Tree is based on data and modelling from the United States so is of limited direct use for international case studies, but could be substituted for regionally appropriate statistics (e.g., [93,95]).

Few alternative approaches to quantifying air quality and pollution were identified in other candidate tools. The Pollution Removal by Vegetation tool follows Jones et al. [93] and uses an atmospheric transport model (EMEP4UK; [96]) to better characterise the spatial and temporal distribution of air pollutants, though is limited to the United Kingdom. Ecoserv-GIS quantifies the relative capacity and demand for air pollution reduction based on land cover characteristics and population density in conjunction with varying buffer distances. The Green Infrastructure Valuation Toolkit, Greenkeeper and InForest each use, or are based upon, the i-Tree air pollution removal method.

### 3.2.5. Erosion, Runoff, and Flood Mitigation

The impacts of trees and vegetation cover on erosion, runoff and flood mitigation are modelled by five of the shortlisted tools (Table 6). These models estimate various components of the water balance, soil loss, and nutrient transport. Both APSIM and i-Tree Eco use point or area-based models and therefore do not explicitly consider spatial dependencies. The Revised Universal Soil Loss Equation (RUSLE; [97]) and related forms are used by most models quantifying soil loss or nutrient transport. The RUSLE predicts annual soil losses by sheet and rill erosion as a function of rainfall erosivity, soil erodibility, topography, land use management and erosion control practices. InVEST and LUCI also include spatial models that route the transport of materials across watersheds and into stream networks or water bodies. This allows for more complex model behaviour that considers both the composition and configuration of land cover types across a study region, but typically will require very fine resolution terrain data and suitable calibration for reliable results [98,99]. Economic valuation is provided by both InVEST (annual water yield) and i-Tree, based on the potential production of hydroelectricity and cost of stormwater treatment in the United States, respectively.

**Table 6.** Summary of the shortlisted tools and associated models that quantify erosion, runoff, and flood mitigation.

| Tool | Model Name | Description | Output Variables |
| --- | --- | --- | --- |
| APSIM | Erosion | Estimated soil erosion using the modified USLE [100]. | Soil loss. |
| APSIM | SoilWater | A daily water balance model based on CERES [101] and PERFECT [102] with additional improvements including allowing for unsaturated flows. | Evapotranspiration, runoff, infiltration, drainage, surface ponding, lateral outflow, etc. |
| ARIES | Soil erosion control | Soil loss and volume avoided by the presence of vegetation, estimated using the RUSLE [97] in conjunction with land cover data. | Soil loss, soil loss avoided by vegetation. |
| InVEST | Annual water yield | Sub-watershed scale water yield, calculated using annual precipitation and the Budyko curve (following [103,104]) | Actual and potential evapotranspiration, water yield, total extraction. |

**Table 6.** *Cont.*

| Tool | Model Name | Description | Output Variables |
|---|---|---|---|
| InVEST | Seasonal water yield | Estimates the relative contribution of pixels to baseflow (during dry weather) and quickflow (during or post-rain). | Relative baseflow, recharge and quickflow. |
| InVEST | Nutrient delivery ratio | Estimates the transport of nitrogen and phosphorus to streams. | Nutrient loads and export by watershed, per-pixel nutrient load reaching streams. |
| InVEST | Sediment delivery ratio | Estimates soil loss using RUSLE [97] and the proportion of sediment reaching streams (following [105]) | Soil loss, sediment exported, sediment deposition and retention. |
| i-Tree Eco | Stream flow and water quality | Estimates components of the water balance using a tree-based ecohydrological model. | Interception, evaporation, transpiration, potential evapotranspiration, avoided runoff. |
| LUCI | Erosion and sediment | Identifies areas with high risk of gully and rill erosion or depositing sediments into nearby water features using the Topographic Wetness Index (TWI; [106]). | Erosion vulnerability risk. |
| LUCI | Flood mitigation | Identifies parts of the landscape where water is likely to accumulate following large rainfall events and characterises features that mitigate flows. | Flood mitigation and interception capacity classes. |
| LUCI | Nitrogen and phosphorus | Estimates nutrient loads and transport using topographic flow routing. | Nutrient loads, accumulated loads, stream and lake concentrations. |
| LUCI | RUSLE | Estimates soil loss and erosion risk using the RUSLE [97]. | Soil loss, erosion risk, sediment delivery risk. |

The erosion, runoff, and flood mitigation models used by each of the shortlisted tools have solid foundations in the scientific literature and most will be suitable for farm-scale applications where high-quality input data is available. Topographic data resolution can have significant impacts upon the results of hydrological flow [98,107] and soil loss models [99,108]. Finer grained data can better represent microtopography that is important for farm scale analyses; however, the optimal resolution is ultimately a balance of data availability, processing times, and intended use of model outputs [109]. The spatial scale, underlying uncertainty, and adequate representation of soil properties, land cover, and weather present additional sources of error, e.g., [110,111]. The nature of these processes also means that they should usually be modelled at watershed scale, which is unlikely to correspond to the farm boundary. Validation using streamflow or water quality observations is critical to modelling these NCBs with confidence and credibility.

Many alternative approaches to quantifying erosion, runoff and flood mitigation have been developed. The RUSLE does not describe several important mechanisms, and models for estimating wind [112], gully [113] and streambank erosion [114] have been developed that could provide a more complete representation of the NCBs provided by agroforestry. Alam and Dutta [115] describe additional nutrient pollution models designed for point-to-catchment scale applications that have been commonly applied to agricultural systems. The Soil and Water Assessment Tool (SWAT; [116]), Catchment Analysis Tool (CAT; [117]), Australian Water Resources Assessment (AWRA; [118,119]) and the Vegetation Optimality Model (VOM; [120]) are biophysical models that quantify and include interactions between water balance and vegetation processes.

3.2.6. Biodiversity

Variables relating to biodiversity are modelled by four shortlisted tools (Table 7). The models variously quantify habitat attributes, indices of species abundance, and habitat suitability for birds. ARIES quantifies forest fragmentation based on post-processing [121]

of the European Space Agency Climate Change Initiative Land Cover (ESA-CCI-LC) products [122] at a spatial resolution of 300 m, and therefore is of limited use for decision making in farm-scale agroforestry. LUCI is designed to maintain connectivity and optimise the establishment of new habitat for species of interest. The wildlife habitat model from i-Tree Eco predicts habitat suitability for 9 bird species based on land cover/use and vegetation characteristics. InVEST includes three separate approaches that quantify mean species abundance, known risks to existing ecosystems, or the relative magnitude of habitat quality, rarity and degradation based on land use change and proximal threats.

**Table 7.** Summary of the shortlisted tools and associated models that quantify measures of biodiversity.

| Tool | Model Name | Description | Output Variables |
|------|-----------|-------------|------------------|
| ARIES | Forest fragmentation | Sourced directly from the global relative magnitude of forest fragmentation dataset based on the entropy-based local indicator of spatial association (ELSA; [121]). | N/A |
| InVEST | Habitat quality | Estimates habitat quality and rarity from information on threats, land use, and land cover mapping. | Relative habitat degradation, quality, and rarity. |
| InVEST | Habitat risk assessment | Assesses risk for species or habitats based on an analysis of exposure to threats and magnitude of consequence. | Habitat and ecosystem specific risk. |
| InVEST | GLOBIO | Estimates the proportional change to the abundance of individual species, relative to the same location in pristine condition, in response to stressors (e.g., land use change, development activity, habitat fragmentation). | Mean species abundance. |
| i-Tree Eco | Wildlife habitat | Predicts habitat suitability for 9 bird species using land use, building cover and vegetation characteristics [123]. | Habitat suitability (0–1). |
| LUCI | Habitat connectivity (BEETLE) | Characterises habitat connectivity using known species habitat, patch size requirements, and dispersal ability using a cost-path technique. | Habitat connectivity classification (e.g., existing habitat, conservation priority, expansion possible, or outside of dispersal range). |

Notes: The forest fragmentation metrics used by ARIES are calculated separately and made available as part of the source dataset. Of the bird species considered by the i-Tree Eco wildlife habitat model, only the common starling (*Sturnus vulgaris*) is present in Australia (non-native).

The models used by each of the shortlisted tools can be feasibly implemented with modest data requirements; however, the methods applied are not always appropriate for farm-scale applications in Australia and are relatively simple compared to what is available in the scientific literature. For example, of the habitat suitability models provided by i-Tree Eco only the common starling (*Sturnus vulgaris*) can be found in Australia, and it is non-native. Individual species may not be a suitable proxy for biodiversity [124,125], and validation is required to ensure the credibility of this assumption even where there are strong similarities in taxonomy and ecological requirements between species [126]. This means that the GLOBIO model [127,128] used by InVEST needs to be implemented with careful consideration of species composition, despite the broader focus on threats to biodiversity. The habitat quality and risk assessment models are conceptually aligned with the historical use of habitat as a proxy for biodiversity [129], Australian state conservation listings [130,131] and the IUCN Red List of Ecosystems [132]. These methods are tractable, but will be sensitive to how habitat classifications reflect species ecological requirements [133]. The LUCI approach is well-suited to scenario analysis and developing spatial optimisation strategies; however, it remains dependent on the assumptions regarding proxy species and habitat classifications.

Biodiversity has been quantified using a broad spectrum of methods in the scientific literature. Two common approaches to quantifying biodiversity are stacked species

distribution models and macroecological models [134]. Species distribution models predict suitability of habitat using the known presence-absence or occurrence records [135], whereas macroecological models predict emergent properties of biodiversity such as species richness [136] or compositional turnover [137]. Both approaches seek to link dependent variables (e.g., habitat suitability, species richness, compositional turnover) to environmental conditions, often using data-driven statistical techniques, and draw upon many of the same data sources (e.g., Global Biodiversity Information Facility, Atlas of Living Australia) that would typically be required to run the GLOBIO model. Building on well-supported ecological principles, frameworks exist to translate patterns of biodiversity into meaningful indicators of biodiversity status and trend (e.g., https://research.csiro.au/macroecologicalmodelling/research-areas/indicators-biodiversity-change/, accessed 24 March 2022, [138]). These can facilitate conservation planning [139], provide estimates of extinction risk in the face of future climate change and land use scenarios [140] and be used to support global biodiversity reporting requirements [141].

Other approaches identified by the remaining candidate tools often used habitat mapping as a proxy for biodiversity, though there were exceptions. The Natural Environment Valuation Online quantifies species richness using known occurrence records for pre-selected species. Many techniques for quantifying fragmentation and connectivity are provided by FRAGSTATS [142] and GuidosToolbox [143]. While useful in some cases (e.g., tracking forest fragmentation; [144]), the calculated metrics are difficult to link to ecological processes [145–147].

### 3.2.7. Crop Pollination

Spatially explicit pollination modules are available in ARIES and InVEST (Table 8). The methods used by both tools have origins in Lonsdorf et al. [148], which is focused on wild bees as pollinators. They rely upon nesting site availability and floral resources as key inputs to estimate pollinator supply. Demand for pollination is quantified as the proportion of crop production attributable to animal pollinators (e.g., [149]). InVEST uses guild-specific attributes to control pollinator behaviour (i.e., active seasons and travel distance). It also incorporates central place foraging theory (based on [150]) such that pollinators will more frequently visit patches with higher quality floral resources given the same travel distance. ARIES implies foraging distance through differences in the spatial resolution of land cover (finer pixel size) and crop type layers (coarser pixel size). It also includes modifiers that increase floral resources near freshwater and accounts for the impacts of temperature and radiation on insect activity levels [151]. Both ARIES and InVEST can provide valuation where commodity prices are available by estimating the proportion of crop yield attributable to pollinators; however, economic measures of pollination services are known to be highly uncertain [152].

**Table 8.** Summary of the shortlisted tools and associated models that quantify crop pollination.

| Tool | Model Name | Description | Output Variables |
|------|-----------|-------------|------------------|
| ARIES | Crop pollination | Calculates indexes of pollinator supply using nesting sites, floral resources, distance to fresh water and pollinator activity, and demand using weighted sum of crop dependencies. | Pollinator supply and demand |
| InVEST | Pollinator abundance | Calculates indexes of pollinator supply and crop yields using habitat suitability (nesting sites and floral resources), and guild attributes (incl. foraging distance) and farm characteristics. | Pollinator supply and relative abundance indices, crop yield index for managed and wild pollinators. |

The methods implemented by both ARIES and InVEST for quantifying pollination are flexible enough to run using global or locally developed datasets; however, there are limitations to how these models can be interpreted. There are many dynamic factors that impact pollination but are not captured in these models. These include actual pollinator

abundance, the impact of stressors such as drought and pesticides, changes in population viability over time, inter and intra-species competition, resource depletion, and preferences for different floral resources. While this information is rarely available and would therefore be difficult to implement, it necessitates the use of relative indices that must be treated carefully without field validation. The spatial scale of analyses is another source of uncertainty. For example, spatial grain of land cover information may be too coarse to effectively represent pollinator habitat, floral resources, and suitable substrates. While these limitations are recognised and there have been ongoing efforts to improve upon these models in the scientific literature (e.g., [153]), data for field validation is limited and unlikely to be available outside of specific case-studies or without data collection in the field.

Ecosystem services modelling tools such as InVEST have been widely used for quantifying crop pollination [154], though distinct approaches have been developed. Both ASSET (https://assist.ceh.ac.uk/, accessed 15 February 2021) and Perennes et al. [154] incorporate species distribution models for bee pollinators and use species richness as a proxy for pollination supply. Co$ting nature [155] uses dry matter productivity on non-croplands in conjunction with simple distance decay functions as a proxy for pollination and pest control. Ecoserv-GIS uses habitat and a distance-based visitation probability to quantify pollination supply and demand, based on Schulp et al. [156].

### 3.2.8. Amenity and Recreation

Amenity and recreation benefits are modelled by three of the shortlisted tools (Table 9). Trees in agricultural landscapes can provide amenity benefits (through improved aesthetics/scenic outlooks) and recreation benefits (through the opportunity for undertaking recreational activities, such as walking and cycling). These can be provided to both private landowners and the public (if the sites are openly accessible). Trees can also have indirect influences on recreation activities, for example, through interactions with water quality (which can influence opportunities for recreational fishing, swimming, or boating), biodiversity (which can influence bird watching and nature viewing), and through providing shading and shelter from extreme weather for recreation activities [157].

**Table 9.** Summary of the shortlisted tools and associated models that quantify amenity and recreation.

| Tool | Model Name | Description | Output Variables |
|---|---|---|---|
| Imagine | N/A | Amenity value of farms engaged in agroforestry. Assumes 10% increase in land value over 5 years. | Adjusted land value ($) |
| InVEST | Scenic quality | Viewsheds are used to quantify and categorise the impact of offshore developments on scenic quality. | Categorical (unaffected/very low/low/medium/high) estimates of scenic quality. |
| InVEST | Visitation: Recreation and Tourism | Linear regression models are used to estimate the key determinants (e.g., natural features, infrastructure, land uses) of visitation rates (user-provided, or collated from public domain geotagged images). | Photo-user-days/visitation rate per year/month. Regression coefficients. |
| SolVES | N/A | Statistical modelling (MaxEnt) of social values acquired from survey data as a function of environmental layers. | Value index (0–10). |

Notes: Amenity values for Imagine are based on the methods described by Mendham [45].

Imagine incorporates amenity values transferred from Polyakov et al. [158], who demonstrated that trees on farms tended to increase the land value. InVEST includes a model for conducting a viewshed analysis which can be used to estimate visual impact (visual disamenity) or visual quality (visual amenity). The model assumes a negative impact on views (due to its framing on offshore infrastructure impacts on coastal scenic beauty), however, it is flexible enough to also estimate positive visual quality benefits. InVEST also includes a model for quantifying the benefits of recreation and tourism. It predicts the relative rate of

visitation across a landscape and estimates the contribution of features of the natural and built environment to visitation rates using a simple linear regression. It also allows for the prediction of visitation rates under alternative or future scenarios. SolVES statistically links a range of social values, including aesthetic and recreation values (plus perceptions of cultural, economic, and historic value amongst others), to spatially explicit environmental layers. Public attitude and preference surveys are used to elicit social value responses and to link those responses to specific locations on a study area map. It then uses Maxent (maximum entropy) modelling to derive a quantitative 10-point, social-values index.

The methods implemented by both InVEST and SolVES for quantifying amenity and recreation are flexible enough to run at different scales; however, there are limitations in the outputs produced and in their interpretation. Neither tool directly estimates the number of visits, or the welfare enjoyed by individuals as a result of having access to the recreation site. Estimating the number of visits made to recreation sites across a broad spatial landscape typically requires large-scale visitor surveys, however, these can be both time-consuming and expensive. InVEST uses a proxy measure of visitation by relying on the relationship between geotagged photographs uploaded to the website Flickr and the number of people who visit a location. While Wood et al. [159] provide evidence that a reasonably good relationship exists, there are also several potential biases in the use of photograph information. For example, photograph locations typically capture visual hotspots, but not areas closer to home or more commonly visited. Different recreational activities may also be more or less suited to taking photographs. There is also a spatial discrepancy inherent in geotagged photograph information as it indicates the position of the photographer and not of the subject. Finally, both visitor presence and the sharing of images to social media are known to be socially biased [160]. For InVEST the spatial scale of analyses is another source of uncertainty; the density of photographs varies spatially, and this has ramifications for the cell-size that can be chosen for analysis. SolVES relies on smaller scale primary survey data and combines this with a value transfer model to enable expansion of the analysis to other locations [49,161]. A discussion of various caveats and problems associated with such transfers, such as the compounding of errors in original study data, and requirements that the transfer location should be both biophysically and socially similar to the original site, can be found in Sherrouse et al. [162] and Semmens et al. [161].

A range of alternative approaches have been developed. Several models predict the potential capacity and demand for amenity and recreation using features of the landscape and accessibility. For example, EcoServ-GIS identifies areas where potential recreation capacity and recreation demand coincide and differ; and ARIES covers several case study areas for both visual amenity and recreation services. Other models combine amenity and recreation capacity and demand information with social media data [163] or mobile phone tracking data [164]. For example, Co$ting nature estimates potential recreation services through linking features of the nature landscape with its accessibility to populations and then combines this with geotagged photograph database similar to InVEST.

Finally, there exists a range of approaches that can be used to estimate monetary values. The travel cost method is commonly used to model the environmental quality of recreational sites along with people's expenditure on travelling to sites to deduce a monetary estimate of the welfare of recreational experiences [165,166]. ORVal [167] uses a sophisticated travel cost-method and a large national-scale recreation visitation survey to estimate visits and value for every open access park, path and beach in England and Wales and combine this information into an interactive map-based web tool.

## 4. Discussion

### 4.1. Identifying Tools That Quantify Natural Capital Benefits of Agroforestry and Shortlisting Those Best Suited to Farm-Scale Applications in Australia

None of the identified tools could quantify all the natural capital benefits that were assessed. In their respective reviews, Luedeling et al. [27] and Kraft et al. [28] found that agroforestry tools had restricted options for quantifying important ecosystem services and

NCBs. Both studies focused on a smaller number of tools (*n* = 6 and *n* = 13, respectively) typically designed for modelling tree-crop interactions and production. Our analyses considered a broader range of tools and modelling strategies. Those that were shortlisted were designed for a range of different purposes and provided a broad range of techniques for quantifying NCBs. Despite this, multiple tools are still currently required to address each of the different NCBs that were assessed. Furthermore, few of the tools identified met our suitability criteria for farm-scale modelling in Australia.

The maximum proportion of tools not meeting any single suitability criterion was just 32%, despite 86% not being shortlisted for any reason. This reflects the range of tools evaluated and demonstrates that the reasons why different tools are limited are variable. The lack of ability to quantify NCBs of agroforestry was most common and was typically associated with tools designed for closely aligned but tangential purposes (e.g., for agroforestry design, economic analyses, recording biodiversity assets, crop and pasture productivity in non-agroforestry context). This was a consequence of considering many potential tools representing diverse methods. A lack of suitability for Australian applications or the need for extensive parameterisation was second most common. This is not surprising given the complexity of process-based agroforestry models and the recognised challenges in extending or calibrating niche models for new species and environments [27,28]. Over one fifth of the tools evaluated were also dependent upon pre-calculated, location-specific datasets (e.g., for the UK, or North America) that would in many cases require significant efforts to develop and apply to new regions. Tool availability and maintenance is a significant challenge, also recognised by Luedeling et al. [27] and Kraft et al. [28], meaning that several impactful tools that would otherwise have been shortlisted were not (e.g., SPIF, EnSym). Unexpectedly, the spatial resolution of the data and model outputs was only rarely a limiting factor. Most tools (93%) were, at least notionally, compatible with farm-scale analyses, though some user intervention may be required (e.g., user-provided data or post-processing).

*4.2. Evaluating the Modelling Capabilities of Shortlisted Tools*

Each of the natural capital benefits that we evaluated were quantified using techniques that vary in their maturity, assumptions, and complexity. There was a trade-off between the generality and complexity of models available. For example, APSIM and FullCAM dynamically allocate and transport carbon between different biomass pools, whereas ARIES and InVEST use much simpler lookup tables to assign carbon stocks. Similarly, InVEST and LUCI incorporate spatial methods for flow routing, yet simplify interactions with vegetation and soil moisture that would otherwise be modelled by APSIM or other more nuanced process-based models. Flow routing is not typically considered in more complex point or area-based models. These kinds of trade-offs mean that there was no single tool identified that could best represent each NCB that was assessed.

Timber production, carbon sequestration, erosion, runoff, and flood mitigation, and crop, pasture and livestock productivity included the most well-developed models among the shortlisted tools. These NCBs are backed by decades of scientific research, with tangible biophysical and/or economic impacts, and have strong links to environmental policy and governance mechanisms (e.g., UN Sustainable Development Goals, [168] Reef 2050 Long-Term Sustainability Plan, [169], UN-REDD). Many quantitative methodologies and data sources have been developed that can often be applied in data-poor environments to support decision-making. A significant advantage of using more complex process-based models such as CABALA or APSIM, however, is coupling of carbon and water cycles in vegetation models. This means that both types of natural capital benefits can be quantified simultaneously and with internal consistency. Despite the modularity and flexibility of tools such as APSIM, these models are not commonly extended to incorporate the broader suite of NCBs (but see for example [29,170]) that may be supported by agroforestry.

Biodiversity, wind, shelter and microclimate, and recreation and amenity were each modelled using a variety of published methods; however, these natural capital benefits also

presented the greatest opportunity for further development. Few of the tools quantified biodiversity well from a macroecological perspective, focusing mostly on individual species or habitat types. There are an increasing number of biodiversity indicators becoming available (e.g., the Biodiversity Indicators Partnership, https://www.bipindicators.net/, accessed 24 March 2022) and tools designed for farm-scale applications are currently under development (e.g., https://looc-b.farm/, accessed 24 March 2022). A key purpose of shelterbelts is to provide a windbreak, yet the shortlisted tools cannot quantify the impacts well under dynamic configurations. Representing the spatial-temporal dynamics (see [68]) of these windbreaks under varying spatial configurations would be particularly useful for developing agroforestry decision support tools that can be applied at scale. The recreation and amenity models are heavily restricted by the availability of suitable survey data and could benefit from the exploration of additional non-linear modelling techniques that can account for variable interactions. For example, data on visitation rates and travel distance would enable monetary valuation of recreation and amenity using methods applied to existing, but regionally specific, tools such as ORVal [167] and NEVO. Implementing or building new approaches to quantifying these natural capital benefits in agroforestry systems would provide significant decision support capabilities.

Further challenges and limitations are likely for quantifying crop pollination and air quality. There are many uncertainties associated with pollination models, necessitating the use of proximal variables with relative indices of potential pollination. These models focus on the potential spatial-temporal distribution of pollinators rather than the process of pollination. Developing centralised, accessible databases on regionally important pollinators (not only bees), including their behaviour, occurrence and preferences (e.g., habitat, crop types), would however further facilitate the use of accessible models such as InVEST. Atmospheric transport and pollution models [96,171] may improve capacity for quantifying potential air quality improvements provided by agroforestry, as existing approaches predominantly rely upon point source information. Efforts to improve air quality modelling is likely better targeted towards larger forestry operations given the relatively small footprint of typical agroforestry plantings.

### 4.3. Key Capability Gaps and Opportunities for Future Development

Few capability gaps emerged that equally affected each of the tools assessed given their diversity, though there were common themes that arose. One key capability gap was the lack of dynamic responses to landscape configuration. Agroforestry is associated with many spatially dependent processes, and it was clear that these are usually not captured well, beyond models designed for quantifying hydrological processes. While biophysical modelling frameworks such as APSIM do allow for various spatial interactions to occur, they are usually simplified (but see Hi-sAFe; [22]). In APSIM for example, competition for light, nutrients and water are controlled by tree, crop, and pasture parameters that assume horizontally uniform conditions. Any regions where spatial interactions occur need to be considered and parameterised explicitly, and the appropriate distances to consider will differ depending on both variables (e.g., light, wind) and site-specific conditions (e.g., tree height, species composition, habitat). Wind speed reductions have been demonstrated with APSIM using distance-based zones of influence; however, the current implementation does not allow for changes in wind direction, interactions with varying spatial configurations and assumes fixed tree porosity. Allowing flow routing, wind speed, shading, and inter-species competition, for example, to dynamically respond to changing site conditions would enable a range of scenario and design-based questions to be explored; however, these are not trivial challenges to solve.

Tool and model complexity, accessibility, regional applicability, and interoperability present additional challenges for quantifying the NCBs of agroforestry. This has been previously recognised [27,28] and is reflected in the range of reasons that arose for excluding tools from the shortlist (see Table A1). Many of the more complex but well-developed models take considerable time investment and/or expertise to use. Furthermore, tools are often

written in different programming languages and are not always well-maintained or made available, limiting their accessibility without considerable additional efforts. The level of complexity required to address user needs will vary by application and spatial-temporal scale (e.g., decision support and risk management, natural capital accounting, scientific research), and therefore there is a need for simplified, accessible models. Reducing the time and resource overheads associated with this complexity and accessibility challenge was among the key motivations for the development of tools such as ARIES and InVEST [43], that have both enabled rapid natural capital and ecosystem services assessments globally. The (relative) simplicity of models used by these tools means that they can often be applied more generally in different locations, adapting to local data availability, or drawing upon global data sources. There are strong arguments for building additional capabilities into more complex modelling frameworks such as APSIM (e.g., to support process understanding, internal consistency and feedback; see [27,28] for more detailed discussion); however, there are significant resource requirements for developing and calibrating new modules. It may be possible to meet the needs of many users and a broader range of NCBs by leveraging the existing suite of models and tools that are available.

Hybridised approaches to quantifying the NCBs of agroforestry systems have the potential to increase development speed and maximise flexibility by matching an appropriate level of complexity with end user requirements. Such concepts have been implemented by several of the tools that were evaluated (e.g., ARIES, InVEST, Imagine, Spatial Planning and Investment Framework); however, accessibility, maintenance and interoperability still present challenges. One way to support hybrid quantification of NCBs would be to use open-source software as the 'glue' to bring together different tools or models on an as-needed basis. There are many advantages to using open-source scripting languages such as R or Python, for example. Both provide vast libraries of well-maintained packages that provide an enormous diversity of modelling capabilities, strong support for many different data types, and active developers building Application Programming Interfaces (APIs) to integrate external software. For example, r3PG [172], 3-PG2Py [173] and APSIMX [174] provide APIs for running the 3-PG forest growth model and APSIM in an R or Python environment. A Python API is available for InVEST, and one is currently under development for Imagine. It is also possible to alternate between these languages by calling R from Python, or vice versa. The flexibility and interoperability provided by open-source scripting languages could facilitate rapid quantification of natural capital benefits, while enhancing accessibility, reducing maintenance overheads, and aligning tools and models with a reduced set of programming languages.

## 5. Conclusions

Many opportunities exist for developing new or improving existing tools that quantify the NCBs of agroforestry systems. Ultimately, the best tool for a given application will depend on end-user requirements, and therefore improving flexibility and interoperability are key to making these resources more accessible. We recommend that scientists, land managers and software developers work together to:

1.  Explore opportunities to build upon and streamline the implementation of tools like APSIM to reduce the resources required to assess NCBs at farm scale;
2.  Build capacity to represent spatially dependent processes that can dynamically adapt to different scenarios and landscape configurations;
3.  Develop and publish high quality spatial surfaces (e.g., productivity under alternative climate and management scenarios, biophysical remote sensing models), at appropriate spatial and temporal scales, to support development of new tools;
4.  Repurpose existing biophysical models where possible to increase development speed and minimise barriers to adoption;
5.  Explore opportunities for integrating observations with process-based models to support monitoring and evaluation of existing agroforestry systems and improve model calibration;

6.  Develop APIs and/or implement tools with widely used open-source scripting languages to promote uptake, enable further development, and to facilitate interoperability; and

7.  Design tools with a level of complexity that is appropriate for the required end use.

A better understanding of the NCBs associated with agroforestry systems and their trade-offs can help to build more resilient and sustainable agricultural enterprises, guide informed policy design, and significantly contribute towards interdisciplinary research on nature-based solutions, natural capital and ecosystem services.

**Author Contributions:** S.B.S., A.P.O. and D.S.M. conceived the ideas and designed the methodology. S.B.S. analysed the data and led writing of the manuscript. All listed authors contributed significantly to the original draft preparation and subsequent revisions. All authors have read and agreed to the published version of the manuscript.

**Funding:** This research is supported by CSIRO, through funding from the Australian Government's National Landcare Program.

**Institutional Review Board Statement:** Not applicable.

**Informed Consent Statement:** Not applicable.

**Acknowledgments:** We would like to thank members of the Smart Farming: Perennial Prosperity team for recommending several of the tools included in the review. We would also like to thank Peter Taylor (CSIRO Data 61) and Chris Ware (CSIRO Land and Water) for providing valuable feedback prior to submission.

**Conflicts of Interest:** The authors declare no conflict of interest.

## Appendix A

**Table A1.** List of tools identified as part of this review and capacity for quantifying selected natural capital benefits. Six screening criteria were used to evaluate the suitability of each tool for detailed evaluation of modelling capabilities: (a) quantifies at least one of the selected natural capital benefits of agroforestry; (b) uses methods or data that are compatible with farm-scale analyses; (c) includes, or allow the user to provide, supporting datasets (where required) that can be applied in Australia; (d) uses methods and models that are suitable for Australian applications, or can be applied without extensive parameterisation; (e) provides functionality not implemented by existing Australian tools; and (f) are currently available for use as open-source software, via collaboration, or by web application.

| Tool | Timber Production and Carbon Sequestration | Crop, Pasture, and Livestock Productivity | Wind, Shelter, and Microclimate | Air Quality and Pollution | Erosion, Runoff, and Flood Mitigation | Biodiversity | Crop Pollination | Amenity and Recreation | Economic Measures | Suitability Criteria not Met | Citation/Link |
|---|---|---|---|---|---|---|---|---|---|---|---|
| APSIM | x | x | x | x | x | | | | x | - | [25,26], https://www.apsim.info/ (accessed 1 September 2021) |
| ARIES (for SEEA explorer) | x | | | | x | x | x | x | x | - | [33], https://aries.integratedmodelling.org (accessed 1 September 2021) |
| Farm Forestry Toolbox | x | | | | | | | | x | - | [44], https://www.farmforestrytoolbox.com/ (accessed 7 September 2021) |
| FullCAM 2020 | x | x | | | | | | | x | - | [41], https://www.industry.gov.au/data-and-publications/full-carbon-accounting-model-fullcam (accessed 5 September 2021) |
| Imagine | x | x | x | | | | | x | x | - | [31,45] |

**Table A1.** *Cont.*

| Tool | Timber Production and Carbon Sequestration | Crop, Pasture, and Livestock Productivity | Wind, Shelter, and Microclimate | Air Quality and Pollution | Erosion, Runoff, and Flood Mitigation | Biodiversity | Crop Pollination | Amenity and Recreation | Economic Measures | Suitability Criteria not Met | Citation/Link |
|---|---|---|---|---|---|---|---|---|---|---|---|
| InVEST | x | | | | x | x | x | x | x | - | [34], https://naturalcapitalproject.stanford.edu/software/invest (accessed 1 September 2021) |
| i-Tree Eco | x | | x | x | x | x | | | x | - | [46], https://www.itreetools.org/tools/i-tree-eco (accessed 1 September 2021)) |
| LUCI | x | | | | x | x | | | | - | [47,48], https://lucitools.org/ (accessed 15 September 2021) |
| SolVES | | | | | | | | x | | - | [49], https://pubs.er.usgs.gov/publication/tm7C25 (accessed 9 October 2021) |
| Agroforestry Design Tool | | | | | | | | | | a | https://www.agroforestryx.com/ (accessed 1 October 2021) |
| ASSET | | | | | x | x | x | | x | b,c,d | https://assist.ceh.ac.uk/asset-assist-scenario-exploration-tool (accessed 20 September 2021) |
| Atlas of Living Australia (ALA) 1 | | | | | | x | | | | a | https://www.ala.org.au/ (accessed 1 October 2021) |
| AusFarm Decision Support Software 2 | | x | | | | | | | x | a | https://doi.org/10.25919/d07h-pr78 (accessed 20 September 2021) |
| B£ST | x | | | | x | x | | x | x | d | https://www.susdrain.org/resources/best.html (accessed 20 September 2021) |
| CMSi Site Management | | | | | | | | | | a | https://www.esdm.co.uk/cmsi-introduction (accessed 20 September 2021) |
| COMP8 | | | | | | | | | | a,f | [175] |
| Co$ting Nature | x | | | | x | x | x | x | x | b | http://www.policysupport.org/costingnature (accessed 22 September 2021) |
| Crop Livestock Enterprise Model (CLEM) | | | | | | | | | x | a | [176]; https://www.apsim.info/clem/Content/Details/Overview.htm (accessed 3 September 2021) |
| Digital Agricultural Services (DAS) 2 | | x | | | | | | | x | a | https://digitalagricultureservices.com/ (accessed 5 September 2021) |
| Decision Support System for Agrotechnology Transfer (DSSAT) 2 | | x | | | | | | | x | a | [65] |
| DynACof | x | x | x | | x | | | | | d | [177] |
| EcoServ-GIS | x | | x | x | | x | x | x | | c | https://www.nature.scot/doc/naturescot-research-report-954-ecoserv-gis-v33-toolkit-mapping-ecosystem-services-gb-scale (accessed 20 September 2021) |
| EcoservR | x | | x | x | x | x | | x | | e | https://ecoservr.github.io/EcoservR/ (accessed 20 September 2021) |
| EnSym 3 | x | x | | | x | x | | | | f | https://ensym.biodiversity.vic.gov.au/cms/ (accessed 1 September 2021) |
| EPIC | x | x | x | x | x | | | | x | d,e | [178] |
| ESAT-A | x | | | | x | x | | | | f | [179] |
| European Forest Information Scenario model (EFISCEN) | x | | | | | | | | | d | [180]; https://efi.int/knowledge/models/efiscen/documentation (accessed 27 September 2021) |
| FarmMap4D | | | | | | | | | | a | https://www.farmmap4d.com.au/ (accessed 1 September 2021) |
| Farm-SAFE 4 | | | | | | | | | x | a | [181]; https://www.agforward.eu/ (accessed 20 September 2021) |

**Table A1.** *Cont.*

| Tool | Timber Production and Carbon Sequestration | Crop, Pasture, and Livestock Productivity | Wind, Shelter, and Microclimate | Air Quality and Pollution | Erosion, Runoff, and Flood Mitigation | Biodiversity | Crop Pollination | Amenity and Recreation | Economic Measures | Suitability Criteria not Met | Citation/Link |
|---|---|---|---|---|---|---|---|---|---|---|---|
| Figured | | | | | | | | | x | a | https://www.figured.com/ (accessed 1 September 2021) |
| FlintPro | x | | | | | | | | | e | https://flintpro.com/ (accessed 1 October 2021) |
| Forecaster | x | | | | | | | | x | d | https://www.scionresearch.com/ services/software-and-applications (accessed 20 September 2021) |
| Forest Investment Framework (FIF) | x | | | | x | x | | x | x | c,d,e | [182] |
| FRAGSTATS [5] | | | | | | x | | | | a | [142] |
| GrassGro [2] | | x | | | | | | | x | a | [50] |
| Green Infrastructure Valuation Toolkit | x | | | x | | x | | x | x | d | https://www.merseyforest.org.uk/ services/gi-val/ (accessed 20 September 2021) |
| Greenkeeper | x | | | x | | x | | x | x | c,d | https://www.greenkeeperuk.co.uk/ the-tool/ (accessed 20 September 2021) |
| GuidosToolbox [3] | | | | | | x | | | | a | [143]; https://ec.europa.eu/jrc/en/ scientific-tool/guidos-toolbox (accessed 2 September 2021) |
| Hi-SAFE | x | x | x | | | | | | | d,e | [22] |
| HyPAR | x | x | x | | | | | | | d | [75] |
| ICBM/N | x | | | | | | | | | d,e,f | [183] |
| InForest | x | | | x | x | x | | | | c | http://inforest.frec.vt.edu/ (accessed 20 September 2021) |
| Integrated Biodiversity Assessment Tool (IBAT) [1] | | | | | | x | | | | a | https://www.ibat-alliance.org/ (accessed 1 September 2021) |
| i-Tree Canopy | | | | | | | | | | a,c | [46]; https://www.itreetools.org/ (accessed 1 September 2021) |
| i-Tree Design | x | | | x | x | | | | x | c | [46]; https://www.itreetools.org/ (accessed 1 September 2021) |
| i-Tree Landscape | x | | | x | x | | | | x | c | [46]; https://www.itreetools.org/ (accessed 1 September 2021) |
| Land Use Trade-Offs (LUTO) Model | x | x | | | | x | | | x | b | [184] |
| LOOC-C | x | | | | | | | | | e | https://looc-c.farm/ (accessed 15 September 2021) |
| MESH | | | | | | | | | | a | https://naturalcapitalproject. stanford.edu/software/mesh; http: //justinandrewjohnson.com/mesh/ (accessed 20 September 2021) |
| NEVO (Natural Environment Valuation Online tool) | x | | | | | x | | x | x | b,c | https: //www.leep.exeter.ac.uk/nevo/ (accessed 12 October 2021) |
| OPAL | | | | | | | | | | a | https://naturalcapitalproject. stanford.edu/software/opal (accessed 20 September 2021) |
| ORVal (Outdoor Recreation Valuation Tool) | | | | | | | | x | x | c | [167]; https: //www.leep.exeter.ac.uk/orval/ (accessed 12 October 2021) |
| Pollution removal by vegetation | | | | x | | | | | | c | https://shiny-apps.ceh.ac.uk/ pollutionremoval/ |
| SBELTS | x | x | x | x | | | | | | d,f | [185] |
| Scenario Planning and Investment Framework Tool (SPIF) | x | | | | x | x | | | x | f | [186] |
| SCUAF | x | x | | | x | | | | | d,e,f | [23] |
| SENCE (Spatial Evidence for Natural Capital Evaluation) | x | | | | | x | | | | c,f | https://www.envsys.co.uk/sence/ (accessed 20 September 2021) |
| Simulateur mulTIdisciplinaire pour les Cultures Standard (STICS) [2] | | x | | | | | | | | a | [66] |

**Table A1.** *Cont.*

| Tool | Timber Production and Carbon Sequestration | Crop, Pasture, and Livestock Productivity | Wind, Shelter, and Microclimate | Air Quality and Pollution | Erosion, Runoff, and Flood Mitigation | Biodiversity | Crop Pollination | Amenity and Recreation | Economic Measures | Suitability Criteria not Met | Citation/Link |
|---|---|---|---|---|---|---|---|---|---|---|---|
| TESSA (Toolkit for Ecosystem Service Site-based Assessment) 6 | x | | | | x | x | | x | x | a | http://tessa.tools/ (accessed 15 September 2021) |
| Viridian HydroloGIS | x | | | x | x | x | | x | x | c,f | https://viridianlogic.com/ (accessed 15 September 2021) |
| WaNuLCAS | x | x | x | | | | | | | d | [21] |
| WIMISA | | x | x | | | | | | | f | [187] |
| Yield-SAFE | x | x | | | | | | | | d | [188] |

[1] The Atlas of Living Australia (ALA) and Integrated Biodiversity Assessment Tool (IBAT) do not quantify the relationship between species occurrences and agroforestry, and therefore were considered not to meet criterion a. [2] AusFarm, DAS, DSSAT, GrassGro and STICS provide productivity modelling capabilities; however, they do not account for interactions with trees and were not considered further. [3] A product licence for EnSym could not be obtained at the time of this review and insufficient documentation was available to evaluate the full range of models applied by the tool, therefore it was not considered for detailed review. [4] Farm-SAFE provides economic modelling capabilities for biophysical SAFE models. [5] FRAGSTATS and GuidosToolBox provide various methods for calculating fragmentation and connectivity using spatial analyses and mathematical morphometry; however, there are no explicit links between these metrics and biodiversity in agroforestry systems, so were not considered for the detailed capability review. [6] TESSA provides guidance documentation for evaluating ecosystem services. It has not been considered for detailed capability review as it does not directly quantify the selected natural capital benefits of agroforestry.

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
