# Peer review of "Digital Tools for Quantifying the Natural Capital Benefits of Agroforestry: A Review"

_land, doi:10.3390/land11101668_

Round 1
Reviewer 1 Report
This paper reviews tools that are available to assess eight different natural capital benefits (NCBs) of agroforestry systems in Australia. The paper is very well written and easy to follow throughout. It provides a beneficial overview of different tools and models available in the literature and then screens them to select a subset that focus on NCBs in agroforestry settings. After reviewing the tools that can potentially be used for each of the 8 NCBs it also discusses limitations and challenges with these tools. There is a good section that summarizes limitations of the tools overall and provides a good perspective on what steps can be taken to improve these tools and make them more effective. While this is helpful and provides some direction for future work, what this paper does not do is provide recommendations for the most effective tool or tools for each of the 8 NCBs that are reviewed. With all the work that was done to search the literature and then run and examine these tools it would be beneficial if the authors would provide these kinds of recommendations for readers who are looking to implement tools now to assess NCBs and not wait years for improvements to be developed.
There are several smaller issues that are highlighted in the attached PDF of the paper that should also be addressed.

Author Response
Thank you for your positive and constructive feedback. You raise an important point with regards to recommendations. We have opted to focus on recommending a pathway to improving the tools themselves, rather than a specific recommendation on which one to use. The variety of end-user requirements that have driven current tool development will also dictate which tool is the best choice for any given application. Some users are likely to need far simpler models than others to meet their needs, and we shouldn’t make models more complex than they need to be. This review therefore provides the resources for users to evaluate these models in the context of their own applications. We’ve made this more explicit in the conclusion, by stating that:
‘Ultimately, the best tool for a given application will depend on end-user requirements, and therefore improving flexibility and interoperability are key to making these resources more accessible.’
We’ve also worked through the attached pdf file and have provided responses to your comments below:
- These items operate at different scale. Flooding is often more localized than erosion, which is a process that occurs over much wider areas. These two are often driven by different processes as well. Is there a reason that they are combined here? It may be that there are few tools with flooding capability so they were added here.
Response: We decided to combine these categories, as LUCI is the only tool that models flood mitigation (and is designed to work at very fine scales, e.g. 5 m digital elevation models) and each incorporate hydrological processes in some capacity (e.g. rainfall erosivity in RUSLE). We recognise that there are other types of processes that drive erosion but are not captured in these models. This is noted in the sections below table 6 (lines 537-540):
‘The RUSLE does not describe several important mechanisms, and models for estimating wind [113], gully [114] and streambank erosion [115] have been developed that could provide a more complete representation of the NCBs provided by agroforestry.’
- Would tools make the general list if they did not address at least one NCB?
Response: Yes. There are several tools that we identified during our search that were included in the general list but did not directly quantify the NCBs we discuss (see Table S1).
- Please provide some clarity on what scale this is. Later around L336 a 250 m grid resolution is deemed to course for farm scale. What cut off was used here for this criteria?
Response: Defining a specific scale cut off is difficult, as there are several potential mechanisms that can be used to apply model outputs across scales. We’ve added further details on the farm-scale criterion (lines 185-188). This section now includes the following:
‘Tools were considered suitable for farm-scale analyses where outputs could be either directly (e.g. high resolution images) or indirectly (e.g. area-weighted scaling) attributed to specific landscape features (e.g. shelterbelts, paddocks).’
- This is not one of the NCBs listed so the authors might consider setting this a part slightly or using some different shading so readers can see this clearly.
Response: Thank you for this suggestion, we have changed the shading of economic measures.
- What about these models makes them well suited to scenario analysis. Can you explain the model attributes that contribute to this statement.
Response: The argument can be made for most models to be useful for scenario analysis, so we have removed this sentence.
- There is no indication that RUSLE is used in i-Tree Eco in the table. Please correct the statement here or amend the table.
Response: Thanks for picking this up. We’ve changed the text to read:
‘The Revised Universal Soil Loss Equation [RUSLE; 98] and related forms are used by most models quantifying soil loss or nutrient transport.’
- Are these 9 North American species?
Response: These species are found in North America, but are also more widely distributed across Europe and South America.
- Can you please provide an idea of what finer and coarser scales mean here. This scale issue comes up several times and is an important parameter in some of the decision made in this work. It would be helpful to have some more detail associated with these statements.
Response: This is an important point, thanks for bringing it up. In this instance we’re talking about differences in the pixel resolution used by the datasets that drive the ARIES model. We’ve made this more explicit by changing this line to read:
‘ARIES implies foraging distance through differences in the spatial resolution of land cover (finer pixel size) and crop type layers (coarser pixel size).’
- scale issue again. If this is resolved above then this will make more sense here.
We’ve addressed the farm-scale definition in response to comment 3 above.
Reviewer 2 Report
This is a very well written, comprehensive and useful review of digital tools that can be used to quantify the natural capital benefits of agroforestry systems, and highlights areas for developing new tools or improving existing ones. As such it makes an important contribution to the field .
There are some minor formatting errors - use of a different font in lines 444-445, 447-449, 454-457, 753-762 and in the references.
And two grammatical errors:
line 690 - Alternative (not alternate)
line 1054 - delete and
Author Response
Thank you for your positive feedback and time taken to identify these issues. We have made the following changes:
- Fonts have been corrected throughout the manuscript.
- Several instances of the word ‘alternate’ have been replaced with the correct and intended meaning ‘alternative’, including on line 690.
- We’ve removed the word ‘and’ from line 1024
Reviewer 3 Report
The manuscript titled " Digital tools for quantifying the natural capital benefits of agroforestry: a review " intends to identify tools that quantify natural capital benefits of agroforestry and shortlist those best suited to farm-scale applications in Australia. Moreover, the manuscript tried to evaluate the modelling capabilities of the shortlisted tools and to identify key capability gaps and opportunities for future development. This review was designed to capture a broad range of tools that can quantify the natural capital benefits of agroforestry systems. Tools designed specifically for agroforestry typically focus on timber and food production, with mechanisms to model tree-agriculture interactions. A distinction was made between digital tools or platforms, stand- alone datasets, bespoke mathematical models, and guidance materials in identifying the tools included in this review.
The research is original; it could be characterized as novel and in my opinion important to the field, it also has an almost appropriate structure, and the language has been used well. In the meanwhile, the manuscript has a nice extent (about 11,570 words) and it is comprehensive, as a review. The tables (9) and the figures (2) make the paper reflect well to the reader. For this reason, paper has a "diversity look", not only tables, not only numbers, not only words.
Please revise the correspondence author details and use the appropriate font size and style. Please complete the left side of the first page such as Citation: Lastname, F.; Lastname, F.; Lastname, F. Title. Land 2022, 11, x. https://doi.org/10.3390/xxxxx - Academic Editor: Firstname Last-name and of course type “Publisher’s Note”, Copyright and the creative common logo with the standard image. Use the template from the journal Land (https://www.mdpi.com/files/word-templates/land-template.dot). Moreover the numbers lines must be on the right side of the manuscript.
The title, I think, is all right. The abstract did not reflect well the findings of this study, and it was not the appropriate length. Please revise the abstract of the manuscript and do not forget abstract need to encourage readers to download the paper. The Abstract needs further work. It is not clear. Abstracts should indicate the research problem/purpose of the research, provide some indication of the design/methodology/approach taken, the findings of the research and its originality/value in terms of its contribution to the international literature. The abstract has a long length (about 308 words). Please, revise the abstract, it must be up to 200 words long, for this reason I would be good to reduce [see: Instructions for Authors / Manuscript Submission Overview / Accepted File Formats - (https://www.mdpi.com/journal/land/instructions#submission or https://www.mdpi.com/files/word-templates/land-template.dot)].
For the Methodology chapter, the research conduct has been tested in several areas of the world, with comparable results and will probably be tested in others. Appropriate references to the methodology included in the already published bibliography but you can put more references, from all over the world. Do not forget, the journal “Land” is international.
The introduction is effective, clear, and well organized; it really introduced and puts into perspective what research is negotiating. Please revise the Introduction of the manuscript and include references which are already exists in bibliography. Moreover, it does not contain a clear formulation and description of the research problem. This makes it difficult for the reader to understand the argumentation. Please insert a clear description and justification of the problem the article deals with. It is advised to revise the Discussion and Conclusion. Both sections should be consistent in terms of Proposal, Problem statement, Results, and of course, future work (as you did, but something more should be said about future work of this study). Your conclusion section is short and does not justice to your work. Make your key contributions, arguments, and findings clearer. You must refer to the literature and previous studies in your discussion section.
More discussion is needed, comparing the results of this work related to attributes with those of other studies. I believe that the conclusions section or discussion should also include the main limitations of this study and incorporate possible policy implications. Something more should be said about practical implications.
Please revise the references of the manuscript and include references which are already exists in bibliography. References must have an appropriate style, for this reason I would be good to change [see: Instructions for Authors / Manuscript Preparation / Back Matter / References: - (https://www.mdpi.com/journal/land/instructions or https://www.mdpi.com/authors/references)]. Do not forget, DOI numbers (Digital Object Identifier) are not mandatory but highly encouraged and make the review easier.
Author Response
Thank you for your feedback. Please see our responses to each of the points raised below.
Please revise the correspondence author details and use the appropriate font size and style. Please complete the left side of the first page such as Citation: Lastname, F.; Lastname, F.; Lastname, F. Title. Land 2022, 11, x. https://doi.org/10.3390/xxxxx - Academic Editor: Firstname Last-name and of course type “Publisher’s Note”, Copyright and the creative common logo with the standard image. Use the template from the journal Land (https://www.mdpi.com/files/word-templates/land-template.dot). Moreover the numbers lines must be on the right side of the manuscript.
Response: The manuscript has now been copied into the Land template. All author details, logos, and details have been included as per this template. Line numbers are now displayed on the right hand side as per the template.
The title, I think, is all right. The abstract did not reflect well the findings of this study, and it was not the appropriate length. Please revise the abstract of the manuscript and do not forget abstract need to encourage readers to download the paper. The Abstract needs further work. It is not clear. Abstracts should indicate the research problem/purpose of the research, provide some indication of the design/methodology/approach taken, the findings of the research and its originality/value in terms of its contribution to the international literature. The abstract has a long length (about 308 words). Please, revise the abstract, it must be up to 200 words long, for this reason I would be good to reduce [see: Instructions for Authors / Manuscript Submission Overview / Accepted File Formats - (https://www.mdpi.com/journal/land/instructions#submission or https://www.mdpi.com/files/word-templates/land-template.dot)].
Response: The abstract has been significantly shortened and better reflects the content of the manuscript.
For the Methodology chapter, the research conduct has been tested in several areas of the world, with comparable results and will probably be tested in others. Appropriate references to the methodology included in the already published bibliography but you can put more references, from all over the world. Do not forget, the journal “Land” is international.
Response: This is a review manuscript that was motivated by the need to understand the variety of digital tools that exist for quantifying natural capital in agroforestry systems, and to identify key capability gaps. The review includes many international tools as are discussed and cited throughout the review but would not be appropriate to include in the methods section. We have structured the manuscript with a methods section to describe the systematic process we went through to identify and evaluate these tools. In essence, this is a review of methods!
The introduction is effective, clear, and well organized; it really introduced and puts into perspective what research is negotiating. Please revise the Introduction of the manuscript and include references which are already exists in bibliography. Moreover, it does not contain a clear formulation and description of the research problem. This makes it difficult for the reader to understand the argumentation. Please insert a clear description and justification of the problem the article deals with. It is advised to revise the Discussion and Conclusion. Both sections should be consistent in terms of Proposal, Problem statement, Results, and of course, future work (as you did, but something more should be said about future work of this study). Your conclusion section is short and does not justice to your work. Make your key contributions, arguments, and findings clearer. You must refer to the literature and previous studies in your discussion section.
Response: Thank you for your compliments on our introduction. We have included the key references required to introduce our topic and state our objectives in the introduction. As a review, the additional references are most appropriately placed in their respective subsections. The research objectives are clearly stated in the final paragraph of the introduction, and each has been provided with its own subsection in the discussion. We have focused our recommendations on how we think that the tools will be best developed in the future, rather than recommending a further review process. Note that we have revised our conclusion to make it clearer based on the comments of other reviewers. We have referred to the key literature in our discussion, which is based on the findings of our review of tools provided in their respective subsections.
More discussion is needed, comparing the results of this work related to attributes with those of other studies. I believe that the conclusions section or discussion should also include the main limitations of this study and incorporate possible policy implications. Something more should be said about practical implications.
Response: As a review manuscript, there are few similar studies against which we can compare. We do often refer to two previous review papers (Luedling et al. 2016; Kraft et al. 2021) that we cite 9 times each in the manuscript. These are the most similar works, and we discuss how the scope and findings of our work differs from theirs in several locations.
The various limitations of the tools that we have reviewed are discussed at length throughout the manuscript (each thematic sub-section, and in the discussion). Furthermore, we detail the constraints of our review in the methodology (e.g. Sept 2021 as the time of review, noting that tools evolve quickly over time, lines 172-174).
We have revised the conclusion to indicate that our recommendations may lead to improved policy design (via improved decision support tools).
Luedeling, E.; Smethurst, P.J.; Baudron, F.; Bayala, J.; Huth, N.I.; van Noordwijk, M.; Ong, C.K.; Mulia, R.; Lusiana, B.; Muthuri, C.; et al. Field-scale modeling of tree–crop interactions: Challenges and development needs. Agricultural Systems 2016, 142, 51-69, doi:https://doi.org/10.1016/j.agsy.2015.11.005.
Kraft, P.; Rezaei, E.E.; Breuer, L.; Ewert, F.; Große-Stoltenberg, A.; Kleinebecker, T.; Seserman, D.-M.; Nendel, C. Modelling Agroforestry’s Contributions to People—A Review of Available Models. Agronomy 2021, 11, doi:10.3390/agronomy11112106.
Please revise the references of the manuscript and include references which are already exists in bibliography. References must have an appropriate style, for this reason I would be good to change [see: Instructions for Authors / Manuscript Preparation / Back Matter / References: - (https://www.mdpi.com/journal/land/instructions or https://www.mdpi.com/authors/references)]. Do not forget, DOI numbers (Digital Object Identifier) are not mandatory but highly encouraged and make the review easier.
Response: Thank you. The references have now been formatted with the MDPI EndNote style.
Reviewer 4 Report
I think this review is very useful, and the authors should be commended for the effort they made in compiling it. However, while reading the manuscript I must say that more than once I felt it was somewhat daunting and lengthy. A major contributor to this are the lengthy descriptions of each tool for each service. These should be in the tables, with the text highlighting the main insights.
I also found it peculiar that the term ecosystem services (ES) is rarely used, and often replaced by “ecosystem benefits” or “nature capital benefits”. The eight “categories” listed in L 195-202 are simply ecosystem services. At the very least, you should clarify how these terms differ from one another. This is partly done in L 138-147, but excluding some terms (ES is absent; hence, the difference between ES and NCB is unclear), and should probably appear earlier in the text. I understand that ES attracts fire, but avoiding/bypassing it while in practice discussing it is not an appropriate solution.
The abstract is too long, in a manner that may deter some potential readers during literature search. I am sure some people might simply TLDR and decide not to read the abstract and hence skip the manuscript (I totally disagree with such a strategy, but the fact is that it exists). The abstract itself sometimes goes into too much detail (for example, I am not sure if the list of 8 services is essential), but also lacks clear conclusion/take-home-message. “Several recommendations are provided” – so please put them in the abstract (not necessarily all of them, and obviously very briefly).
A major drawback is that the sources of data to be used in each tool are not specified. While for some ES I assume that remote sensing can suffice, for at least some services (e.g., biodiversity and recreation) there must be some other sources. For example, how can a model predict potential tourist visitations without knowing what is the tourism potential (which may be completely different for farms near and away from a major city, all other things being equal), or how can it predict species richness and pollination potential without field surveys?
L 84-137: Since most/some of these tools are discussed in far more detail below, I think that for the Introduction, it is more important to highlight needs and challenges, rather than presenting each tool. To me, this section read like a list of acronyms and short statements that are loosely connected.
L 147-157: This should probably be the foundations for rewriting L 84-137.
L 225: Are the “seven key co-benefit categories” the eight ES listed in L 195-202?
L 253-262: Looks a little fuzzy
Tables 2-9: Since many tools share output variables, it would be helpful if each variable is a different column. This makes comparisons far easier than in text format.
Author Response
Thank you for your valuable feedback. We have provided some responses, noting that the editor has suggested that we focus on the other 3 reviews.
We’ve used the term natural capital benefit, largely due to conflicts in the definition of ecosystem services. Many consider ecosystem services to only include things that humans directly derive benefit from, and therefore the supporting and intermediate services we cover do not strictly meet that definition. We do associate ecosystem services with natural capital benefits in describing at the beginning of our methods section, so it has not been bypassed entirely.
While the manuscript is long, we believe that this length is justified given the number and complexity of natural capital benefits that were assessed, and the fact that this is a review paper. We submitted this manuscript as a review article with Land, where there are no strict word limits. With regards to the table designs, there are too many differences in the output variables to give each their own column, and many more (order of magnitude in many places) input variables that would otherwise have to be described. We had originally included input and output variables, but it became far too cumbersome, particularly given the current length of the manuscript (as you also note!). On input data, it is incredibly variable in what users can provide. The number of tools that we evaluated meant that we had to make some simplifications, both for readability and practicality.
We have corrected the error on line 225, thank you. The manuscript originally had 7 NCBs that were subsequently expanded to 8.
Round 2
Reviewer 4 Report
I am satisfied with the revisions. Good job!